# Effect of Fly Ash on the Mechanical Properties and Microstructure of Cement-Stabilized Materials with 100% Recycled Mixed Aggregates

**Tao Meng [1],\*, Dawang Dai [1], Xiufen Yang [1] and Hongming Yu [1,2]**

1 College of Civil Engineering and Architecture, Zhejiang University, 866 Yuhangtang Road, Xihu District, Hangzhou 310058, China; 22012231@zju.edu.cn (D.D.); 22012059@zju.edu.cn (X.Y.); yuhongming@zju.edu.cn (H.Y.)

2 Hangzhou Xincheng Dinghong Real Estate Development Co., Ltd., 841 Moganshan Road, Gongshu District, Hangzhou 310005, China

\* Correspondence: taomeng@zju.edu.cn

**Abstract:** The use of recycled mixed aggregates (RMA) in cement-stabilized materials (CSM) is an effective way to dispose of and reuse demolition waste. However, this approach faces various challenges; for example, the drying shrinkage of CSM with 100% RMA is very high, which is unfavorable for use in road engineering. In order to use a simple method to reduce the drying shrinkage of the CSM with 100% RMA and give it reliable strength, the effect of fly ash on the mechanical properties, drying shrinkage, and abrasion resistance of CSM with 100% RMA was investigated in this study, and the mechanism was examined by X-ray Diffraction (XRD), Mercury Intrusion Porosimetry (MIP), and Scanning Electron Microscopy (SEM). The results revealed that the addition of fly ash would decrease the drying shrinkage of CSM with 100% RMA. Moreover, when the amount of fly ash was less than 20%, the later strength increased remarkably despite the slight decrease in the early unconfined compressive strength, indirect tensile strength, compressive and splitting elastic modulus, and abrasion resistance of CSM with 100% RMA. The microstructure analysis results indicated that fly ash increased the decline range of diffraction intensity of $C_2S$ and $C_3S$ at a later age and also helped to optimize the pore structure. Research results of this article can be used to optimize the mechanical properties of CSM with 100% RMA and guide its application in road base.

**Keywords:** fly ash; microstructure; recycled cement-stabilized materials; mechanical properties; recycled aggregate





## 1. Introduction

As the process of urbanization continues to accelerate, many old buildings have been demolished or repaired and consequently a huge amount of construction and demolition waste has been produced [1]. At present, most construction and demolition waste is either temporarily stacked, landfilled, or processed into low-value building materials, such as recycled bricks, and the resource utilization is less than 10% [2,3]. Using construction and demolition waste instead of natural sand to make recycled mixed aggregates (RMA), which can be used as cement-stabilized materials (CSM) in road base, can alleviate the shortage of natural sand, reduce construction costs, and effectively solve the difficulties of construction demolition waste recycling [4].

The composition of construction demolition waste is relatively complex. The main components of RMA made by crushing and screening include concrete, bricks, mortar, glass, and wood chips [5–7], which have poor performance [8–11]. Cachim [12] compared to the performance of RMA and natural aggregates and found that there were few spherical or square particles in the RMA, and the shape index of the RMA was more than 130%

higher than that of natural aggregates. This is not conducive to close combination with the concrete mixture and affects the workability of concrete [13–15]. Zhou Tang et al. [16] found that it was more frequent for the recycled concrete that cracks passed through the aggregate particles in comparison with the natural aggregate. Pitarch [17] used waste tiles, red clay bricks, and RMA sanitary ware as raw materials to partially replace aggregates. The results revealed that the density of natural aggregates was higher than that of RMA, and the water absorption rate varied greatly depending on the material and particle size, ranging from 0.69% to 18.31%. The properties of CSM change depending on the aggregate type. Davis et al. [18] studied the effect of four different aggregates (i.e., mica, limestone, diabase, and granite) on the unconfined compressive strength of CSM. The results indicated that the relationship between the unconfined compressive strength of CSM and the amount of cement changes depending on the aggregate type. In addition, the physical properties of recycled aggregates not only depend on the type of recycled materials, but also on the manufacturing process [19]. Khatib and Otsuki [20,21] claimed that, due to their poor properties, recycled aggregates weaken the interface between the aggregate and the cement paste, resulting in a decrease in the mechanical properties of concrete, resulting lower elastic modulus, flexural strength and splitting tensile strength [22,23].

To improve the performance of recycled aggregate concrete, Xuan [24–26] conducted studies on CSM with RMA. They optimized the composition of CSM with RMA with a response surface methodology and observed that the unconfined compressive strength and elastic modulus of CSM increased with a smaller amount of bricks, and the tensile strength of CSM without brick aggregate was twice that of CSM with 100% brick. Researchers have also attempted to modify CSM by adding active powder or fiber. For example, Messala [27] used woolen fibers, glass fibers, and steel fibers to investigate the effects of different fiber types on the properties of recycled aggregate concrete, and the study indicated that the addition of fibers can effectively optimize the mechanical properties of recycled aggregate concrete. This enhancement is mainly due to these fibers limit the generation of micro-cracks and development of crack width in concrete. Baricevic [28] combines human-made steel fibers with unsorted waste tires and recycled steel fibers to develop an environmentally and economically hybrid fiber-reinforced concrete; this could save up to 33% per m$^3$ of concrete without compromising the material properties. Li [29] investigated the influence of alkali-activated fly ash on the mechanical properties of construction waste and explored the mechanism of this effect by X-ray Diffraction (XRD), Scanning Electron Microscopy (SEM) and Fourier Transform Infrared Spectrometry (FTIR). The results indicated that the material internal structure becomes more compact and compressive strength of the material can be significantly improved when the alkali-activated fly ash in incorporated. This is because the material produced by alkali activation has a higher cementing capacity. Tangchirapat et al. [30] improved recycled aggregate concrete properties by high fineness of fly ash. The results showed that the slump loss behavior could be improved when the replacement of fly ash is about 35~50% in recycled aggregate concretes. Lei [31] found that the durability of recycled concrete can be significantly enhanced by modifying recycled concrete with PVA emulsion or 1.5% nano-SiO$_2$ solution, and replacing cement with 10% fly ash.

In this paper, RMA and low-cost fly ash are used to prepare modified CSM with 100% RMA, and the influence of fly ash on the mechanical properties, drying shrinkage, and abrasion resistance of this CSM are investigated. The mechanism of action is explored by XRD, Mercury Intrusion Porosimetry (MIP), and SEM to examine the effects of fly ash on the phase composition, pore structure distribution, and micromorphology of CSM with 100% RMA. This study provides a simple and easy method to reduce the drying shrinkage of CSM with 100% RMA, which will promote the application of this CSM in actual projects.

## 2. Experiment

### 2.1. Materials

The materials used were as follows:

1. Ordinary Portland Cement was provided by China Hangzhou Fuyang Qianchao Cement Co., Ltd., and it met the P·O 42.5 standard of General Portland Cement in China, and the main components are listed in Table 1.
2. Fly ash, of which the technical performance indicators met the Use of Fly Ash in Cement and Concrete in China requirements. Its density was 2350 kg/m$^3$, and its main chemical composition is presented in Table 1.
3. Recycled mixed aggregates were provided by China Hangzhou Qianjiang New City Government Garden Construction Co., Ltd. It mainly contained concrete rocks, crushed bricks, ceramic tiles, and wood chips, which accounted for 61.3%, 31.7%, 6.6%, and 0.4% of the total weight, respectively. The particle size specifications were 0–9.5 mm (fine particle size) and 9.5–31.5 mm (coarse particle size), and the aggregate gradation is displayed in Figure 1, meeting the requirements of the Specifications for Design of Highway Asphalt Pavement (in China).

**Table 1.** Chemical composition of cement and fly ash (wt %).

| Composition | Al$_2$O$_3$ | SiO$_2$ | CaO | Fe$_2$O$_3$ | SO$_3$ | MgO | Free-CaO | K$_2$O | LOI |
|---|---|---|---|---|---|---|---|---|---|
| Cement | 4.36 | 22.37 | 61.08 | 3.38 | 2.45 | 2.43 | 0.68 | - | 0.8% |
| Fly ash | 26.18 | 41.83 | 3.41 | 3.26 | - | 0.61 | - | 1.31 | 5.6% |

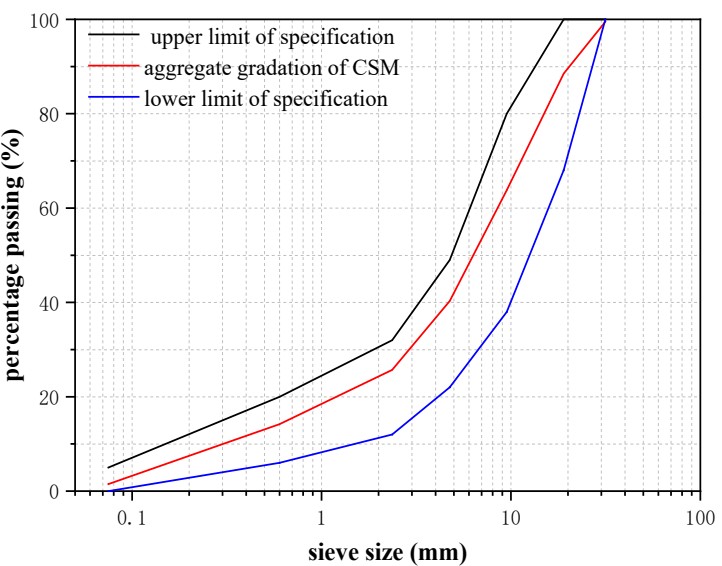

**Figure 1.** Aggregate gradation of RMA.

### 2.2. Mix Ratio

Table 2 presents the mixing ratio of CSM with 100% RMA under different fly ash replacement rates (i.e., 0%, 10%, 20%, 30%, and 40%), where the content as a proportion of the mass of cement was 6% and water was 15.8% (according to the Marshall compaction test).

**Table 2.** The mixing ratio of CSM with 100% RMA under different fly ash replacement rates (wt %).

| Specimens | RMA | Cement | Fly Ash | Water |
|---|---|---|---|---|
| CF0 | 82.1 | 4.93 | 0 | 12.97 |
| CF1 | 82.1 | 4.44 | 0.49 | 12.97 |
| CF2 | 82.1 | 3.94 | 0.99 | 12.97 |
| CF3 | 82.1 | 3.45 | 1.48 | 12.97 |
| CF4 | 82.1 | 2.96 | 1.97 | 12.97 |

*2.3. Experimental Method*

2.3.1. Method of Sample Preparing and Curing

Following the specification of Standard JTG E51-2009 [32], the RMA, cement, and water were stirred together for 3 min. Then, the stirred mixture was poured into steel molds and compacted to form Φ100 mm × H100 mm cylinder specimens for strength tests, Φ150 mm × H150 mm cylinder specimens for abrasion resistance tests and 100 mm × 100 mm × 400 mm cuboid specimens for drying shrinkage tests. The mixture was compacted to 3 MPa for 1 min with an automatic compactor after being placed in the mold. The cylinder specimens were cured for 7, 14, 28 and 180 d, while the cuboid specimens were cured for 7 d, under room conditions at 20 °C temperature and 95% relative humidity.

2.3.2. Test Methods for Mechanics, Deformation, and Durability

1. Unconfined compressive strength (UCS) and indirect tensile strength (ITS): The Φ100 mm × H100 mm cylinder specimens were removed from the curing room after curing for 7 d, 14 d, and 28 d. Additionally, 9 specimens were tested at each age for both UCS and ITS tests according to JTG E51-2009 [32]. The loading rate was approximately 1 mm/min. During the loading process, the vertical and horizontal displacements of specimens were recorded by an extensometer.

2. Compressive elastic modulus: The Φ100 mm × H100 mm cylinder specimens were removed from the curing room after curing for 180 d, and 9 specimens were tested for compressive elastic modulus according to JTG E51-2009 [32]. The predetermined load was divided into five levels. After each load level was applied for 1 min, it was unloaded, allowing the specimens to recover the elastic deformation. The operation was then repeated step by step, and the loading rate was set to 1 mm/min.

3. Splitting elastic modulus: The Φ100 mm × H100 mm cylinder specimens were removed from the curing room after curing for 180 d and 15 specimens were tested for splitting elastic modulus according to JTG E51-2009 [32]. The predetermined load was divided into six levels. After each load level was applied for 1 min, it was unloaded, allowing the specimens to recover the elastic deformation. The operation was then repeated step by step, and the loading rate was set to 1 mm/min.

4. Drying shrinkage properties: Three cuboid specimens for each group were measured by dial gauge and electronic scale for drying shrinkage deformation and weight loss of each specimen within 31 days. The calculation formula is as follows:

$$\text{Shrinkage strain}: \ \varepsilon_i = \delta_i / l; \tag{1}$$

$$\text{Drying shrinkage coefficient}: \ \alpha_{di} = \varepsilon_i / w_i; \tag{2}$$

$$\text{Total shrinkage coefficient}: \ \sum \varepsilon_i / \sum w_i. \tag{3}$$

$w_i$ represents water loss rate;
$\delta_i$ represents drying shrinkage;
$i$ represents the *i*-th measured data;
$l$ represents the initial length of the specimen.

5. Abrasion resistance: The Φ150 mm × H150 mm cylinder specimens were removed from the curing room after curing for the 28 days and 3 specimens were tested for abrasion resistance tests according to JTG E51-2009 [32]. The specimens were fixed in a scouring barrel and water was poured into it with 5 mm above the surface of the specimen, and a 10-Hz frequency was used to flush them for 30 min. The mass loss of the specimens was then measured.

2.3.3. Test Methods for Microstructure

1. Phase composition analysis—XRD: The specimens for XRD analysis were broken into tiny fragments after the UCS tests were performed, and particles of cement mortar were

picked and ground to a residue ratio on the 80 μm sieve less than 2% for a specific curing time. A Bruker D8 ADVACNCE X-ray diffractometer from Bruker Company, Karlsruhe, Germany was used for the XRD analysis. A CuKa radiation source, scanning angles of 10°–80°, a step length of 0.026° and an acceleration voltage of 40 kV were applied in this experiment.

2. Pore structure analysis—MIP: The micropore structure of CSM with RMA was determined by an MIP test. The specimens for MIP were broken into small fragments and dried at 60 °C for 24 h after performing UCS. The mercury analyzer used for this experiment was an AutoPore IV9510 automatic mercury porosimeter (McMurraytic Instruments Co., Ltd., Norcross, GA, USA).

3. Microscopic morphology and composition analysis—SEM: The specimens used for SEM were disintegrated into small fragments and the small fragments were sprayed with gold for 60 s in an SBC-12 small particles sputtering apparatus to make the specimens conductive. The microscopic structure of the specimens under different magnifications was observed with a FEG-650 Quanta field emission scanning electron microscope (ThermoFisher Scientific Company, Waltham, MA, USA).

## 3. Results and Discussion

### 3.1. Influence of Fly Ash on the Mechanical Properties of CSM with 100% RMA

#### 3.1.1. Unconfined Compressive Strength

Figure 2 presents (a) UCS of the CSM with 100% RMA under different fly ash replacement rates and (b) the ratio of UCS at different ages to the 7 d strength. Figure 2a indicates that, as the replacement rate of fly ash increased, the UCS of CSM with 100% RMA gradually decreased. When the replacement rate of fly ash was ≤20%, the addition of fly ash had little effect on the UCS of CSM with 100% RMA. However, when the replacement rate of fly ash was >20%, the incorporation of fly ash significantly reduced the UCS.

To study the influence of age on the UCS of the CSM, the 7 d strength of each group was used as the benchmark (100%), and the ratios of the 14 d and 28 d strengths to the 7 d strength were calculated, as illustrated in Figure 2b. It can be seen that, with an increase in age, the USC of each group of the CSM gradually increased. Specifically, the incorporation of fly ash greatly increased the strength later, which demonstrates that adding fly ash is beneficial to the later USC development of the CSM.

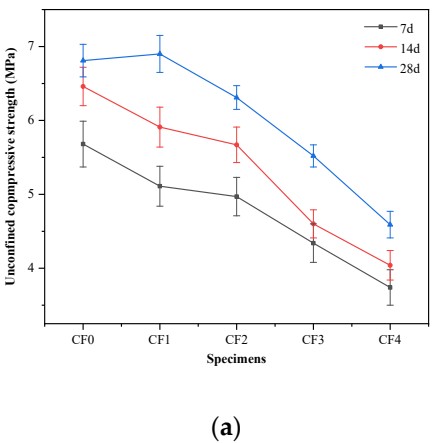

(**a**)

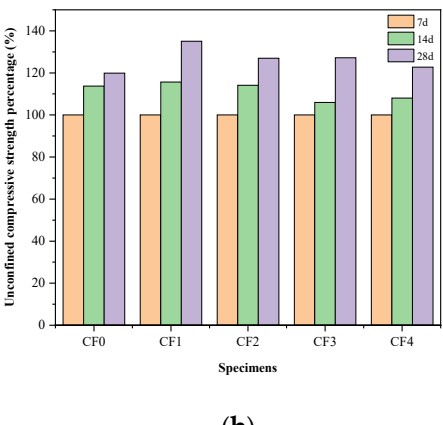

(**b**)

**Figure 2.** (**a**) UCS of the CSM with 100% RMA under different fly ash replacement rates and (**b**) the ratio of UCS at different ages to the 7 d strength.

#### 3.1.2. Indirect Tensile Strength

Figure 3 illustrates (a) ITS of CSM with 100% RMA under different fly ash replacement rates and (b) the ratio of ITS at different ages to the 7 d strength. It can be seen from Figure 3a that, as the replacement rate of fly ash increased, the ITS of the CSM gradually

decreased. When the replacement rate of fly ash was ≤20%, the ITS decreased slightly, whereas when the replacement rate of fly ash was >20%, the ITS decreased significantly.

To study the influence of age on the ITS of the CSM, the 7 d ITS of each group was used as the benchmark (100%), and ratios of the 14 d and 28 d strengths to the 7 d strength were calculated, as illustrated in Figure 3b. It can be seen that with an increase in age, the ITS of the CSM gradually increased. The 14 d ITS of specimens CF0, CF1, CF2, CF3, and CF4 was higher than that of the 7 d indirect tensile strength, exhibiting an increase in 125%, 132%, 131.3%, 150%, and 223.5%, respectively. The 28 d ITS of the specimens was higher than the 7 d ITS by 130.4%, 146%, 145.8%, 200%, and 247%, respectively. Thus, an increased replacement rate of fly ash increased the ITS in the later period, indicating that the addition of fly ash is beneficial to the development of the later ITS of the CSM.

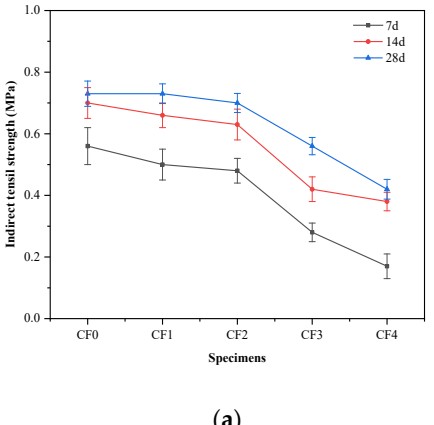

(**a**)

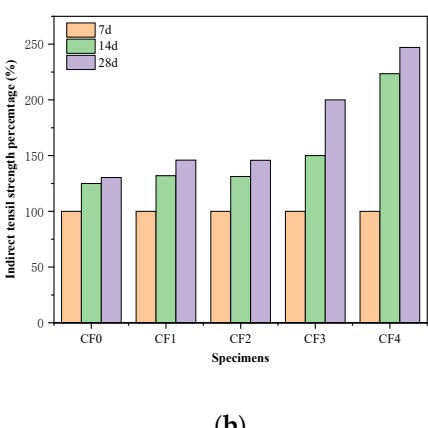

(**b**)

**Figure 3.** (**a**) ITS of CSM with 100% RMA under different fly ash replacement rates and (**b**) The ratio of ITS at different ages to the 7 d strength.

### 3.1.3. Compressive and Splitting Elastic Modulus

Figure 4 illustrates the compressive and splitting elastic modulus of CSM with 100% RMA under different fly ash replacement rates, where CF0 was used as a baseline. It can be seen that when the replacement rate of fly ash was 10%, 20%, 30%, and 40%, compared to the baseline, the compressive elastic modulus of the CSM was reduced by 9.8%, 25.1%, 36.2%, and 38.0%, respectively, while the splitting elastic modulus was reduced by 17.7%, 31.5%, 44.2%, and 51.4%, respectively. This indicates that the incorporation of fly ash can significantly reduce the compressive and splitting elastic modulus of the CSM.

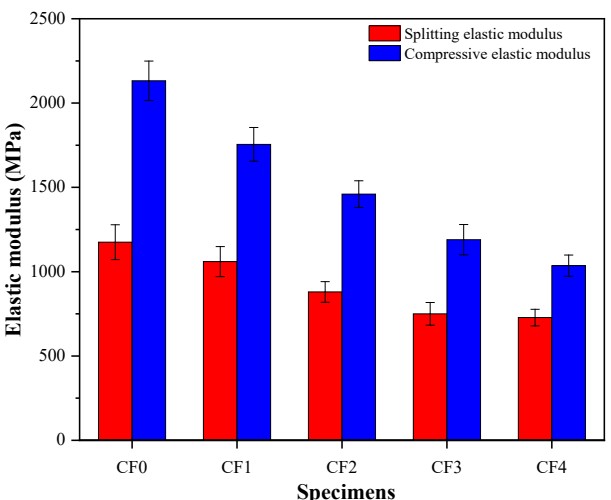

**Figure 4.** The compressive and splitting elastic modulus of CSM with 100% RMA under different fly ash replacement rates.

### 3.2. Influence of Fly Ash Replacement Rate on Drying Shrinkage and Deformability of CSM with 100% RMA

Figure 5 presents the changes in (a) drying shrinkage strain and (b) water loss rate of CSM with 100% RMA over time under different fly ash replacement rates. The results indicate that as the age increased, the drying shrinkage strain and water loss rate of the CSM continued to increase. The 31 d drying shrinkage strain of specimens CF0, CF1, CF2, CF3, and CF4 was 833 με, 817 με, 740 με, 630 με, and 525 με, respectively, and the water loss rate was 12.65%, 13.68%, 13.94%, 14.56%, and 15.34%, respectively. These results indicate that with an increase in the replacement rate of fly ash, the dying shrinkage strain of the CSM gradually decreases while the water loss rate gradually increases. This may be because the specific surface area of fly ash is large and its secondary hydration reaction leads to an increase in water demand, which makes the water loss rate of CSM with 100% RMA gradually increase with an increase in the fly ash replacement rate. In addition, fly ash optimized the internal structure of the CSM by refining large holes, thereby improving the drying shrinkage performance of the CSM by the filling effect and secondary hydration reaction.

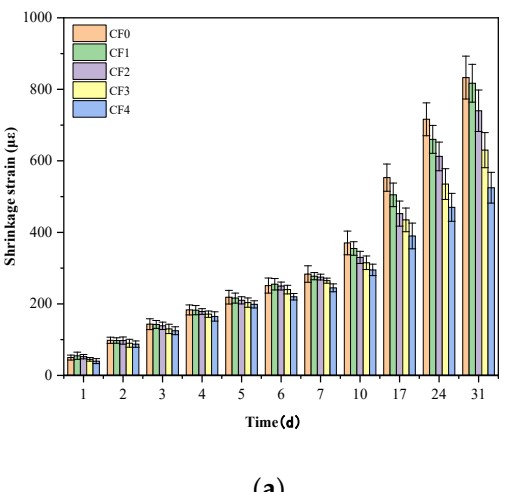

(**a**)

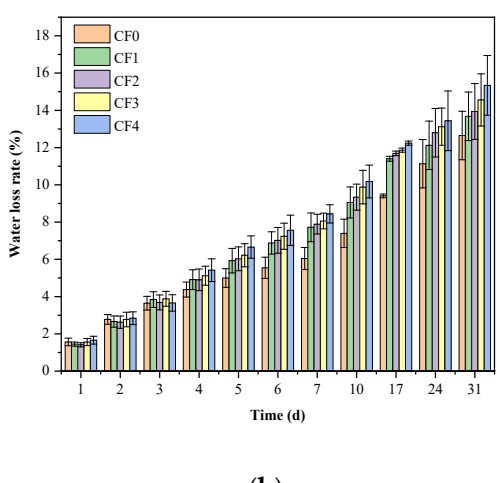

(**b**)

**Figure 5.** Changes in (**a**) shrinkage strain and (**b**) water loss rate of CSM with 100% RMA over time under different fly ash replacement rates.

Figure 6 presents the total drying shrinkage coefficient of CSM with 100% RMA under different fly ash replacement rates. It can be seen that the total shrinkage coefficient of specimens CF0, CF1, CF2, CF3, and CF4 at an age of 31 d was 65.84 με/%, 59.72 με/%, 53.06 με/%, 43.27 με/%, and 34.22 με/%, respectively. The higher the replacement rate of fly ash, the lower the total shrinkage coefficient. When the replacement rate of fly ash was 10%, 20%, 30%, and 40%, the total drying shrinkage coefficient was reduced by 9.3%, 19.4%, 34.3%, and 48.03%, respectively, compared to the baseline group (CF0). The test results indicate that the incorporation of fly ash can significantly reduce the total drying shrinkage coefficient of the CSM and can significantly improve the ability of the CSM to resist drying shrinkage.

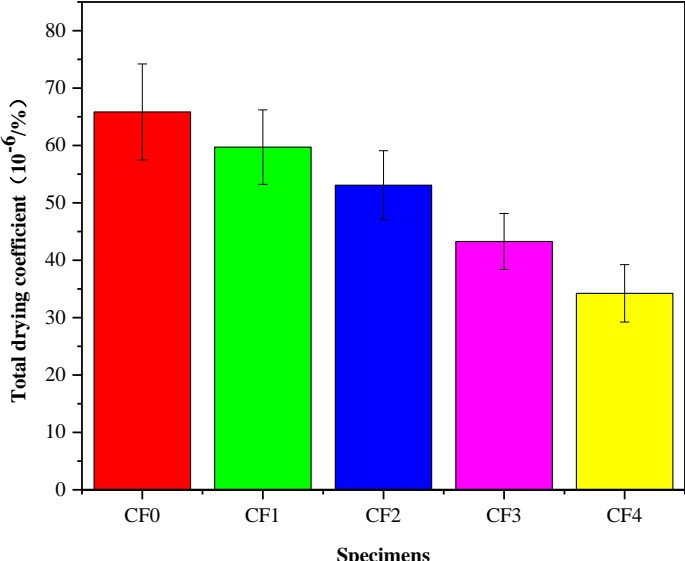

**Figure 6.** Total drying shrinkage coefficient of CSM with 100% RMA under different fly ash replacement rates.

### 3.3. Influence of Fly Ash Replacement Rate on Abrasion Resistance of CSM with 100% RMA

Figure 7 illustrates the mass loss rate of CSM with 100% RMA under different fly ash replacement rates. It can be seen that the mass loss rate of specimens CF0, CF1, CF2, CF3, and CF4 was 0.094%, 0.101%, 0.118%, 0.132%, and 0.149%, respectively. The higher the replacement rate of fly ash, the higher the mass loss rate. When the replacement rate of fly ash was 10%, 20%, 30%, and 40%, the mass loss rate was increased by 7.4%, 25.5%, 40.4%, and 58.5%, respectively, compared with the baseline, indicating that the incorporation of fly ash can reduce the abrasion resistance of the CSM.

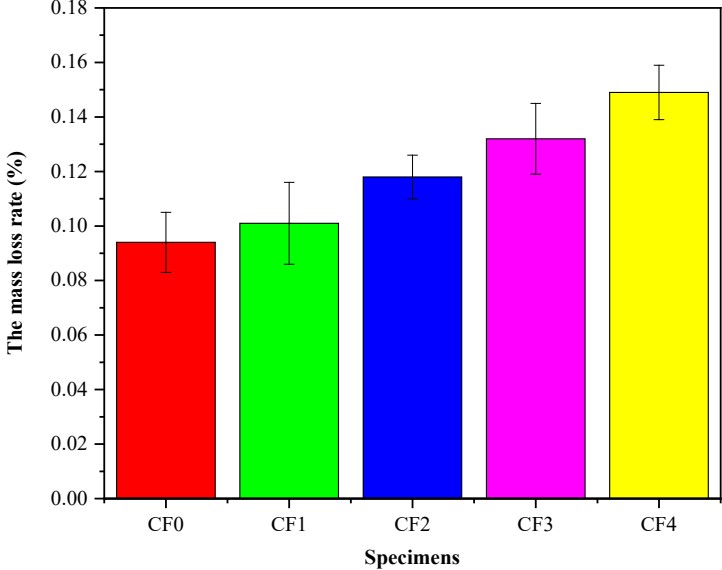

**Figure 7.** The mass loss rate of CSM with 100% RMA under different fly ash replacement rates.

### 3.4. Effect of Fly Ash Replacement Rate on the Microstructure of CSM with 100% RMA and Mechanism Analysis

3.4.1. XRD Mineral Composition Analysis

Figure 8 presents the XRD pattern analysis results of CSM with 100% RMA under different fly ash replacement rates. The main phase composition of the CSM included quartz ($SiO_2$), limestone ($CaCO_3$), dicalcium silicate ($C_2S$), tricalcium silicate ($C_3S$), hydrated

calcium silicate gel (C-S-H), and ettringite (AFt). To compare and analyze the influence of the fly ash replacement rate on the XRD phase of the CSM, the characteristic peaks of $C_2S$ and $C_3S$ at $2\theta = 32.2°$ and $32.6°$, and the characteristic peaks of $CaCO_3$ and C-S-H at $2\theta = 29.4°$ were selected. As illustrated in Figure 8, at the age of 28 d, the characteristic peak area of $C_2S$, $C_3S$, $CaCO_3$, and C-S-H of the baseline group (CF0) was the largest. After adding fly ash, the characteristic peak area of $C_2S$ and $C_3S$ of groups CF1, CF2, CF3, and CF4 gradually decreased. Although C-S-H is difficult to detect, the dispersion peak of C-S-H can be found. The characteristic peaks of $CaCO_3$ and C-S-H also slightly decreased, indicating that as the replacement rate of fly ash increased, cement clinker minerals decreased, and the resulting C-S-H gel also decreased, thus explaining the reason for the compressive and tensile strength reduction.

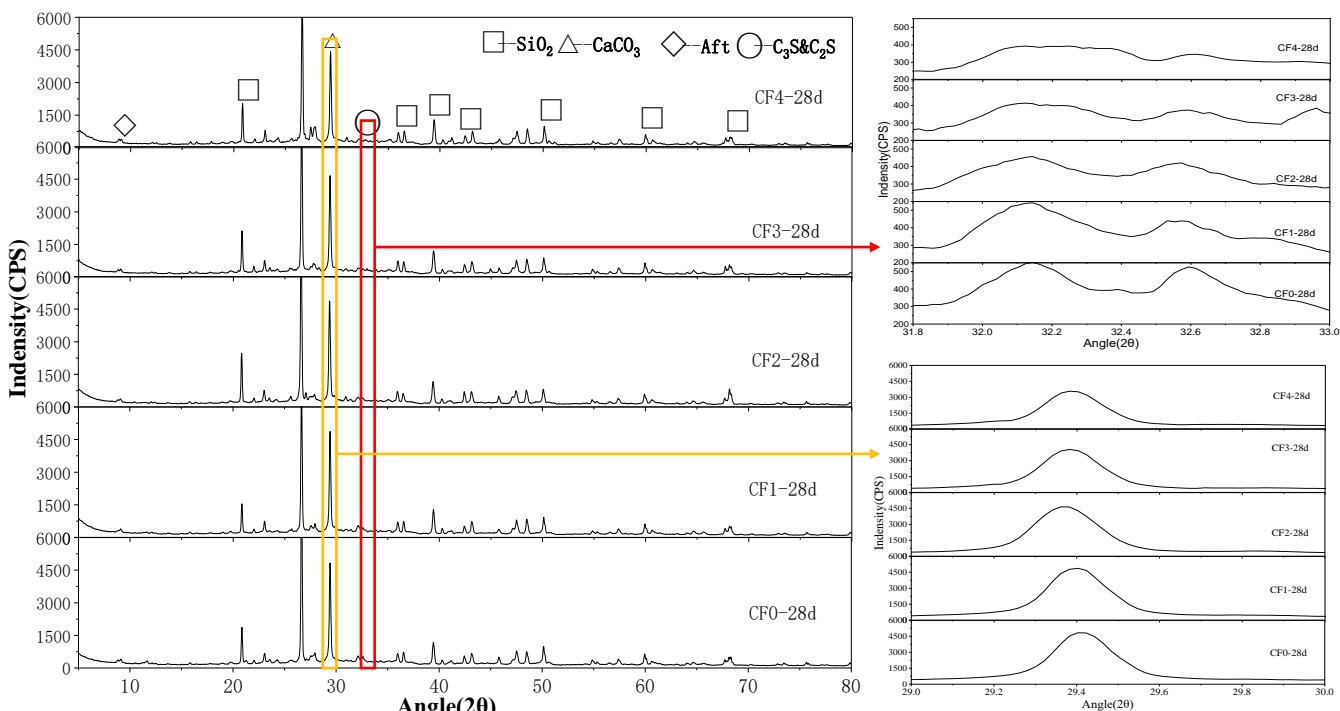

**Figure 8.** X-ray diffraction pattern analysis results of fly-ash-modified CSM with 100% RMA and a partial enlarged view of characteristic peaks at an age of 28 d.

The semi-quantitative method was further used to calculate the peak intensities of $C_2S$, $C_3S$, and C-S-H (excluding $CaCO_3$ of the CF0 group) of the fly-ash-modified CSM with 100% RMA at different ages. The calculation results are presented in Figure 9. It can be seen that with an increase in age, the $C_2S$ and $C_3S$ peak intensities of groups CF0, CF1, CF2, CF3, and CF4 gradually decreased, while the C-S-H peak intensities gradually increased. The main reason is that the cement clinker minerals $C_2S$ and $C_3S$ in the CSM were continuously consumed with the increase in age and gradually hydrated to form C-S-H gel. At the same age, with an increase in the fly ash replacement rate, the $C_2S$, $C_3S$, and C-S-H peak strengths of the CSM tended to gradually decrease. This is because part of the cement was replaced by fly ash. The higher the fly ash replacement rate, the fewer cement minerals contained in the stabilized material, and the fewer cement hydration products at the same age.

In terms of the macroscopic performance, the strength of the CSM gradually decreased as the replacement rate of fly ash increased. The $C_2S$ and $C_3S$ peak intensity of groups CF1, CF2, CF3, and CF4 at an age of 7 days was 213.21, 225.40, 187.11, and 172.48, respectively, which was 12%, 6.9%, 22.7%, and 28.8% lower than the 242.21 peak intensity of the baseline group (CF0), respectively. When the replacement rate of fly ash was >20%, the $C_2S$, $C_3S$, and C-S-H peak strength of the CSM was significantly reduced. This may be the reason for

the reduction in the compressive and tensile strength of the CSM when the replacement rate of fly ash was >20%.

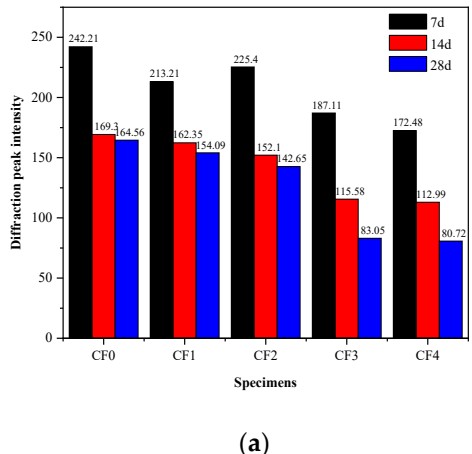

(**a**)

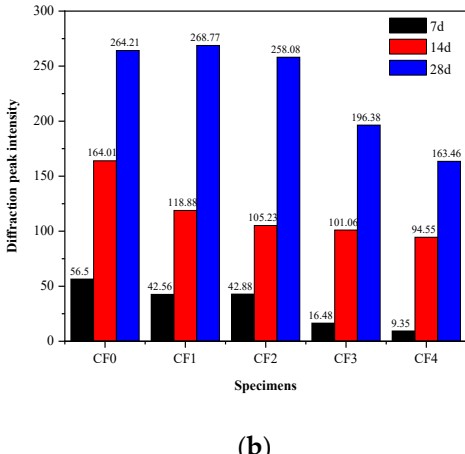

(**b**)

**Figure 9.** (**a**) $C_2S$ and $C_3S$ and (**b**) C-S-H peak intensity of fly-ash-modified CSM with 100% RMA at different ages.

### 3.4.2. MIP Pore Structure Analysis

Figure 10 presents the pore structure data of fly-ash-modified CSM with 100% RMA at an age of 7 d and 28 d. The results indicate that the porosity of groups CF0, CF1, CF2, CF3, and CF4 at 7 d was 32.42%, 31.19%, 30.8%, 29.81%, and 30.15%, respectively. As the replacement rate of fly ash increased, the porosity shows the trend of first decreasing and then increasing. At 28 d, the average pore size of groups CF0, CF1, CF2, CF3, and CF4 was 27.6 nm, 26 nm, 25.7 nm, 29.1 nm, and 32 nm, respectively, displaying the trend of first decreasing and then increasing. This indicates that, when the replacement rate of fly ash is low, it is beneficial to reduce the porosity and average pore size of the CSM. The reason is that when the fly ash content is low, it can fill the voids between the aggregates to make the CSM denser and optimize the pore structure. However, the addition of fly ash is not conducive to the hydration of cement. When the amount of fly ash is too high, the cement hydration products will decrease, which will help the formation of the internal pore structure of the CSM. By comparing and analyzing the pore structure data of different ages, it can be seen that with increasing age, the average pore size and porosity of the CSM were significantly reduced. This indicates that with increasing age, cement and fly ash continue to hydrate, and the hydration products gradually fill the pores in the CSM, thereby optimizing the porosity and average pore size of the CSM.

According to Zhongwei's classification of microscopic pores in cement-based materials into four levels according to pore size [33], the effect of the replacement rate of fly ash on the pore structure distribution of the CSM was further compared and analyzed. Figure 10 indicates that, at an age of 7 d, the proportion of harmful pores (>200 nm) in groups CF0, CF1, CF2, CF3, and CF4 was 54.32%, 46.49%, 43.33%, 42.82%, and 47.22%, respectively, while the proportion of non-hazardous pores (<20 nm) was 20.69%, 20.72%, 23.14%, 20.67%, and 18.38%, respectively. The proportion of harmful pores gradually decreased and the proportion of non-hazardous pores first increased and then decreased with an increasing fly ash replacement rate. This indicates that the incorporation of fly ash can reduce the proportion of large pores in the CSM and optimize its pore structure distribution.

Comparative analysis of the pore structure distribution of fly-ash-modified CSM with 100% RMA at 7 d and 28 d indicates that, with an increase in age, the proportion of harmful pores (>200 nm) of the fly-ash-modified CSM with 100% RMA decreased while the proportion of harmless pores (<20 nm) increased. For example, in the CF1 group, the proportion of harmful pores (>200 nm) decreased from 46.49% at 7 d to 35.2% at 28 d, while the proportion of harmless pores (<20 nm) increased from 20.72% at 7 d to 25.14% at 28 d.

This occurred because the C-S-H gel produced by the continuous hydration of fly ash and cement gradually filled the pores of the CSM.

The main reason for the drying shrinkage deformation of CSM is the evaporation and loss of internal moisture. Fly ash can reduce the porosity and average pore size of the CSM, and moisture inside the CSM cannot easily evaporate. Furthermore, fly ash can significantly optimize the pore structure distribution, reduce the proportion of harmful pores where water can easily evaporate, and increase the proportion of small pores that are harmless and where it is more difficult for water to evaporate. Therefore, the incorporation of fly ash can effectively improve the drying shrinkage properties of the CSM.

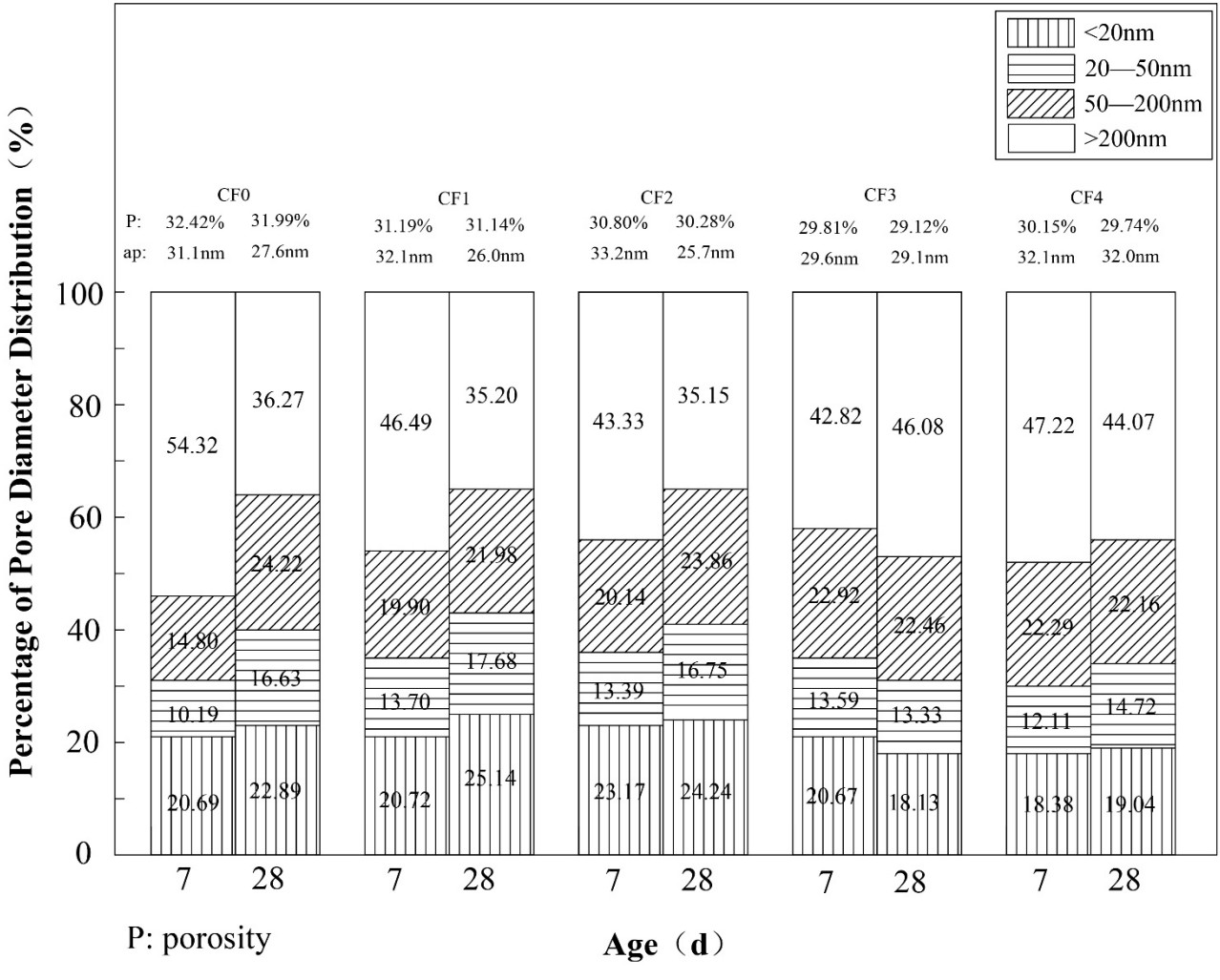

**Figure 10.** Distribution of pore structure of fly-ash-modified CSM with 100% RMA.

### 3.4.3. SEM Microscopic Morphology Analysis

Figure 11 presents the microstructure diagram (×1000) of CSM with 100% RMA at 7 d under different fly ash replacement rates. It can be seen that when the replacement rate of fly ash increased, more spherical fly ash particles filled the aggregates. The baseline group (CF0), had a greater number of pores, and the pores were mostly large in diameter. With an increased fly ash replacement rate, the CF1, CF2, and CF3 specimens exhibited fewer pores, and large pores were significantly reduced, with mostly small pores remaining. This result is consistent with the aforementioned MIP pore structure analysis results.

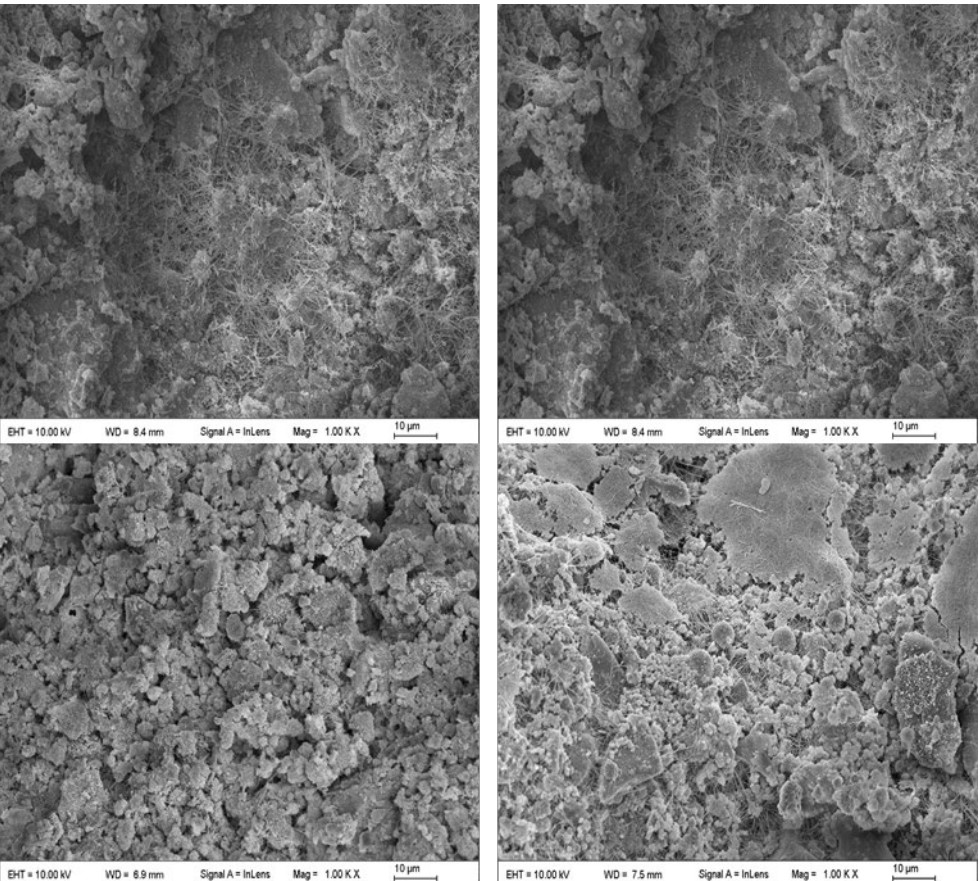

**Figure 11.** Microstructure diagram of CSM with 100% RMA at 7 d under different fly ash replacement rates (×1000). (Note: the upper left is CF0, the upper right is CF1, the lower left is CF2, the lower right is CF3).

Figure 12 displays the micromorphology of CF1 at different ages. It can be seen that the spherical particles were fly ash, while the needle-like and flocculated substances were cement hydration products. Comparative analysis of the microscopic morphology of CSM at 3 d, 7 d, 14 d, and 28 d indicates that in the early period of 3 d and 7 d, there were more cement hydration products, and the fly ash particles were still smooth and spherical. This indicates that the cement had begun to hydrate in the early stage; however, the fly ash had not yet undergone a hydration reaction, and the hydration rate was relatively low. As the age increased, the fly ash particles gradually began to hydrate from the surface. At 28 d, the fly ash particle ball began to collapse, and intersected with the surrounding cement hydration products, filling the pores of the CSM.

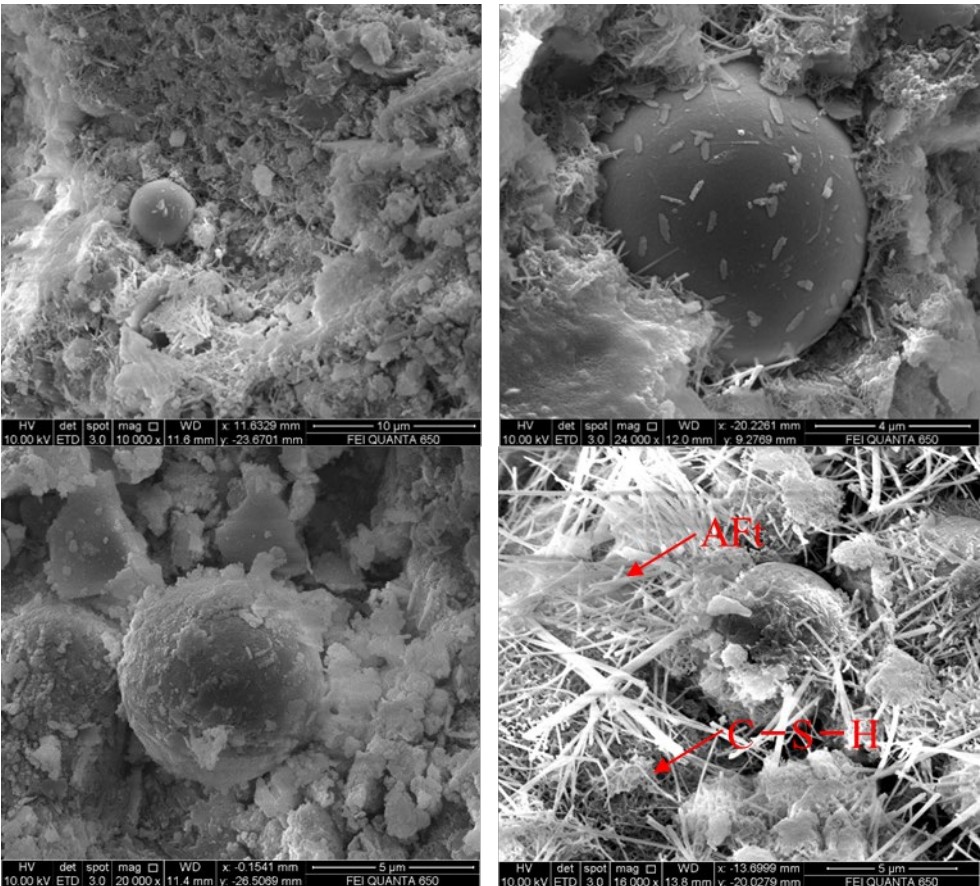

**Figure 12.** Micromorphology of fly-ash-modified CSM with 100% RMA at different ages (CF1 group). (Note: the upper left is 3 d, the upper right is 7 d, the lower left is 14 d, the lower right is 28d).

## 4. Conclusions

This study examined the influence of fly ash on the mechanical properties, drying shrinkage deformation, and abrasion resistance of CSM with 100% RMA, and explored the mechanism of action by XRD, MIP, SEM, and other microscopic methods. The main conclusions are as follows:

1. The incorporation of fly ash can reduce the unconfined compressive strength, indirect tensile strength compressive and splitting tensile rebound modulus of CSM with 100% RMA at all ages. However, it is benefit to the later strength development as the incorporation of fly ash. Moreover, When the replacement rate of fly ash was ≤20%, fly ash had little effect on the strength of CSM with 100% RMA. When the replacement rate of fly ash was greater than 20%, the strength of the CSM was significantly reduced.

2. The incorporation of fly ash can significantly improve the drying shrinkage performance, increase the mass loss, and reduce abrasion resistance of CSM with 100% RMA. When the replacement rate of fly ash was 10%, 20%, 30%, and 40%, the total drying shrinkage coefficient was reduced by 9.3%, 19.4%, 34.3%, and 63.21%, respectively, compared to baseline (0%).

3. XRD phase analysis results revealed that the incorporation of fly ash reduced the early cement clinker mineral consumption and C-S-H gel generation of CSM with 100% RMA. However, with an increase in age, fly ash continued to hydrate, and the later cement clinker mineral consumption and C-S-H gel production gradually increased, explaining the macroscopic reason for the greater increase in the later strength of the fly-ash-modified CSM with 100% RMA.

4. MIP pore structure analysis and SEM results indicated that fly ash can reduce the porosity and average pore size of CSM with 100% RMA and optimize the pore

structure distribution by filling the gaps between the aggregates. The proportion of harmful pores in CSM with 100% RMA gradually decreased while the proportion of harmless pores gradually increased, which may be the reason for the improved drying shrinkage performance of CSM with 100% RMA.

There are many modification methods for CSM with 100% RMA. In this study, low-cost, simple and feasible fly ash was used to improve the SCM performance. However, there are still many performance modification methods that can be tried, such as blending of fibers and nanomaterials.

**Author Contributions:** Data curation, D.D.; investigation, X.Y.; project administration, T.M.; supervision, T.M.; validation, T.M. and H.Y.; visualization, H.Y.; writing—original draft, D.D.; writing—review and editing, D.D. and X.Y. All authors have read and agreed to the published version of the manuscript.

**Funding:** This work was supported by the Zhejiang Provincial Key Research and Development Program (2020C04013) and the National Natural Science Foundation of China (52078453).

**Data Availability Statement:** All data, models, and code generated or used during the study appear in the published article.

**Acknowledgments:** The authors are grateful to the reviewers for their helpful advice and comments.

**Conflicts of Interest:** The authors declare no conflict of interest.

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
