# Peer review of "Effect of Fly Ash on the Mechanical Properties and Microstructure of Cement-Stabilized Materials with 100% Recycled Mixed Aggregates"

_minerals, doi:10.3390/min11090992_

Round 1
Reviewer 1 Report
The manuscript entitled: “Effect of fly ash on the mechanical properties and microstructure of cement-stabilized materials with 100% recycled mixed aggregates” is in line with the Minerals journal. It based on original research. The article is well organized, however it requires minor changes:
- Authors: please be coherent with template.
- Abstract: please develop all abbreviations, when are the first time in the text, including: XRD, MIP, and SEM.
- Keywords: please add “recycled aggregate”.
- Introduction: lines 50-52 – could you be more accurate about the numbers?
- Introduction: lines 59, 63 and others – please unify the references, give consequently only surname or first name and surname in all article.
- Introduction: line 75 – “Researchers” – please specify.
- Introduction: line 95 - please develop all abbreviations, when are the first time in the text: XRD, SEM and FTIR.
- Materials: please define type of cement in lines 115-117.
- Point 2.3.1. and 2.3.2. – please add the number of prepared specimens for each test.
- Discussion: lack of discussion and comparison achieved results with literature (IMPORTANT!).
- Please add information about authors contribution.
Reviewer 2 Report
The paper investigates the effects of cement substitution with fly ash in cement-stabilized materials. The manuscript is well written, and the topic is timely and of interest to the academic and professional communities. The paper in this form can be published in Minerals.
Author Response
Thank you for your comments!
Reviewer 3 Report
The article is devoted to solving a global problem - the disposal of large-scale industrial waste. This problem is of concern to a large number of scientists around the world. The article examines a wide range of characteristics of the studied CSM material, which contains 2 types of industrial waste: recycled mixed aggregates and fly ash. The article presents interesting scientific results and relationships. However, below is the following list of comments and recommendations that authors should consider:
- [line 118]. What does it mean «level II fly ash»? It should be explained in the article
- [line 128]. What does the symbol «f-CaO» means? Perhaps, it is the same that CaOfree. Please, correct it in Table 1 according to the generally accepted abbreviation
- [line 129]. What the "specification" the authors are referred to in Fig. 1. This specification should be indicated in the article.
- [line 149]. What was the time period the specimen was cured before UCS test? Just the abstract phrase «specific age» is mentioned in the article.
- [line 246]. How the shrinkage coefficient was determined?
- [line 305]. It should explain, why porosity for maximum replacement of fly ash (CF4 - 40 %) is increased, whereas, in contrast, the replacement of cement by fly ash from 0 to 30 % has a tendency to decrease in porosity.
- [line 319]. There should probably be indicated Figure 10 instead of Figure 9.
- Why the microstructure for CF1 specimen is absent (Figure 11)? It should be presented.
- Microstructure for which specimen is presented in Figure 12?
- Short resume according to data of SEM analysis is recommended to add into the Conclusion section.
Reviewer 4 Report
In abstract "the drying shrinkage of CSM with100% RMA is very large" change large with "high"
Avoid to use abbreviation in abstract such as XRD, SEMÄ°, etc. Use full name and then abbreviation
i advice to the authrs to check the below papers for sustainability, various application areas of fly ash
1- Zhou H, Bhattarai R, Li Y, Li S and Fan Y 2019 Utilization of coal fly and bottom ash pellet for phosphorus adsorption: Sustainable management and evaluation Resources, Conservation and Recycling 149 372–80
2-Bhagath Singh G V P and Subramaniam K V L 2019 Production and characterization of low-energy Portland composite cement from post-industrial waste Journal of Cleaner Production 239 118024
3- R.D. Hooton, Current developments and future needs in standards for cementitious materials, Cement Concrete Res. 78 (2015) 165-177.
4-Aydin, E. “Staple wire-reinforced high-volume fly-ash cement paste composites”, Construction and Building Materials, Volume 153, 30 October 2017, p.393-401.
add space between the text and referecnes. correct typo line 138. please check them all
provide section numbering for all description of the tests
more contribution is expected for authors, please provide novel discussion for the experimental results one example "Thus, an increased 219
replacement rate of fly ash increased the ITS in the later period, indicating that the addition of fly ash is beneficial to the development of the later ITS of such CSM" this is commonly accepted known fact. There are many for the other section. Please proide more better explanation by considering the microstructure.
please show all observed phases in SEM graphs such ash calcium silicate hydrate, calcium hydroxide, aluminum hydrates, etc.
recommendation section is missing and must be added
correct typo in line 142 "tests. And the"
Round 2
Reviewer 3 Report
Some of the following comments were not taken into account by the authors:
- [line 256]. It should be explaining, what does the term «dry shrinkage coefficient» and «total shrinkage coefficient» mean. How the dry shrinkage coefficient and total shrinkage coefficient was determined/calculated?
- In Figure 6, the vertical axis represents the "Total drying factor" parameter. However, when describing Figure 6, the text does not mention the "Total drying factor" parameter.
- [line 314]. It should explain, why porosity for maximum replacement of fly ash (CF4 - 40 %) is increased, whereas, in contrast, the replacement of cement by fly ash from 0 to 30 % has a tendency to decrease in porosity.
Reviewer 4 Report
authors did some of the comments and suggestions
Author Response
Thank you so much for your comment! I modified and improved the article according to your suggestions to make this article more in line with the requirements. You have helped me a lot, thank you sincerely!
Round 3
Reviewer 3 Report
All comments have been corrected.
The article is recommended for publication